# Twisted photonic Weyl meta-crystals and aperiodic Fermi arc scattering

Hanyu Wang[1,2,3,7], Wei Xu[1,2,3,7], Zeyong Wei[4,7], Yiyuan Wang[5,7], Zhanshan Wang[4], Xinbin Cheng [4], Qinghua Guo [6] ✉, Jinhui Shi [5] ✉, Zhihong Zhu [1,2,3] ✉ & Biao Yang [1,2,3] ✉

As a milestone in the exploration of topological physics, Fermi arcs bridging Weyl points have been extensively studied. Weyl points, as are Fermi arcs, are believed to be only stable when preserving translation symmetry. However, no experimental observation of aperiodic Fermi arcs has been reported so far. Here, we continuously twist a bi-block Weyl meta-crystal and experimentally observe the twisted Fermi arc reconstruction. Although both the Weyl meta-crystals individually preserve translational symmetry, continuous twisting operation leads to the aperiodic hybridization and scattering of Fermi arcs on the interface, which is found to be determined by the singular total reflection around Weyl points. Our work unveils the aperiodic scattering of Fermi arcs and opens the door to continuously manipulating Fermi arcs.

Fermi arcs bridging Weyl points offer insights into topological semi-metal phases[1–10] and exotic phenomena such as chiral anomaly[11,12] and nontrivial surface states[13–15]. Very recently, manipulating Fermi arcs has further stimulated more interests, e.g., possible transport properties in junctions of Weyl semimetals[16], quantized chiral magnetic current[17], and Fermi arc reconstruction in synthetic photonic lattice[18], etc. Understanding, constructing and manipulating Fermi arcs[19–21] can lead to the development of novel electronic/photonic devices and quantum technologies, just like the reconstruction of Fermi loop surface states, which have been cleverly demonstrated in magnetically tunable 3D photonic crystals[22] with the closed topological surface states extending across the surface Brillouin zone in the forms of torus knots.

With the booming progress in two-dimensional materials, stacking and twisting have become simple and elegant routes toward studying topological physics[23–28]. A number of interesting effects have been observed in the twisted systems[28–33]. Twisted bi-layer lattices can be classified into two categories, that are commensurate and incommensurate lattices[34–36]. These commensurate angles guarantee translation symmetry and each angle constructs a new superlattice. For those incommensurate angles, arbitrary lattice configuration is possible. Twisting physics also paves an unprecedented way towards manipulating Fermi arcs, e.g., Fermi arc reconstruction[37–39], arcless surface states[37,40], Lifshitz transition[41–44], and Weyl orbit[4,45], etc. Most of these previous studies[17,37–40] focus on the system preserving translation symmetry as they can be well described by Bloch band theory. However, Fermi arcs in non-translational systems[46,47] deserve further study as the interaction between Fermi arcs and non-translation would fundamentally expand our understanding of solid-state physics and crystallography.

## Results

Here, we develop a twisted bi-block Weyl meta-crystal and study the continuously twisted photonic Fermi arcs on its interface. It is worth mentioning that only the interface exhibits aperiodic features, while both the Weyl meta-crystals individually preserve translational symmetry. As the aperiodic counterpart of conventional Fermi arcs, the

[1]College of Advanced Interdisciplinary Studies, National University of Defense Technology, Changsha 410073, China. [2]Hunan Provincial Key Laboratory of Novel Nano-Optoelectronic Information Materials and Devices, National University of Defense Technology, Changsha 410073, China. [3]Nanhu Laser Laboratory, National University of Defense Technology, Changsha 410073, China. [4]Institute of Precision Optical Engineering, School of Physics Science and Engineering, Tongji University, Shanghai 200092, China. [5]Key Laboratory of In-Fiber Integrated Optics of Ministry of Education, College of Physics and Optoelectronic Engineering, Harbin Engineering University, Harbin 150001, China. [6]School of Physics and Electronics, Hunan University, Changsha 410082, China. [7]These authors contributed equally: Hanyu Wang, Wei Xu, Zeyong Wei, Yiyuan Wang. ✉e-mail: guoqh@hnu.edu.cn; shijinhui@hrbeu.edu.cn; zzhwcx@163.com; yangbiaocam@nudt.edu.cn

twisted Fermi arcs (refers to the interface Fermi arcs between two Weyl semimetals rotated from each other) exhibit a denser and more complex pattern in momentum space. We found that at small twist angles, the meta-crystal can be regarded as a homogenous medium (i.e., metamaterials) because the quasi-period is much larger than the Weyl wavelength (the wavelength at the Weyl points), which exhibits negligible periodic modulations. For large angles, the quasi-period is comparably as large as the Weyl wavelength[48], and the scattering between quasi-period tends to dominate the reconstruction of Fermi arcs in twisted bi-block Weyl meta-crystals. From the round-trip reflection phase between the two Weyl meta-crystals, i.e., periodically distributed metamaterials in spatial space, we explored the twisted Fermi arcs that bridge aperiodic Weyl points. Finally, based on the feasibility of the bi-block meta-crystals, we experimentally observed the continuously twisted Fermi arcs.

We assume that the top Weyl block is fixed while the bottom block clockwise rotates a relative angle $\theta$. When $\theta = 0$, the two blocks form a

whole ideal Weyl meta-crystal, and actually there is no interface. The interface could be created after slightly introducing a dielectric gap. Then arcless surface states appear as shown in Fig. 1a[37,40], where the net topological charge is vanishing on the interface. When $\theta \neq 0$, the projected Weyl points on the interface are redistributed, and a pair of twisted Fermi arcs occurs there (see Fig. 1b). Imposing periodic conditions makes both Weyl points and Fermi arcs periodically distributed (Fig. 1c) and densely scattered back and forwards after further twisting (Fig. 1d). One may consider the interface Fermi arcs arising from the hybridization of two fully decoupled Fermi arcs, e.g., from Fig. 1e (with $h \rightarrow \infty$) to Fig. 1f (with $h \rightarrow 0$).

We first show the way to solve interface Fermi arcs of homogeneous Weyl metamaterials (see section I in Supplementary Information and Supplementary Fig. 1) and then consider the periodic modulation in Weyl meta-crystals. Metamaterials imply homogeneous media, i.e., there is no spatial-periodic modulation. From metamaterials, we can calculate and predict twisted Femi arcs for each arbitrary

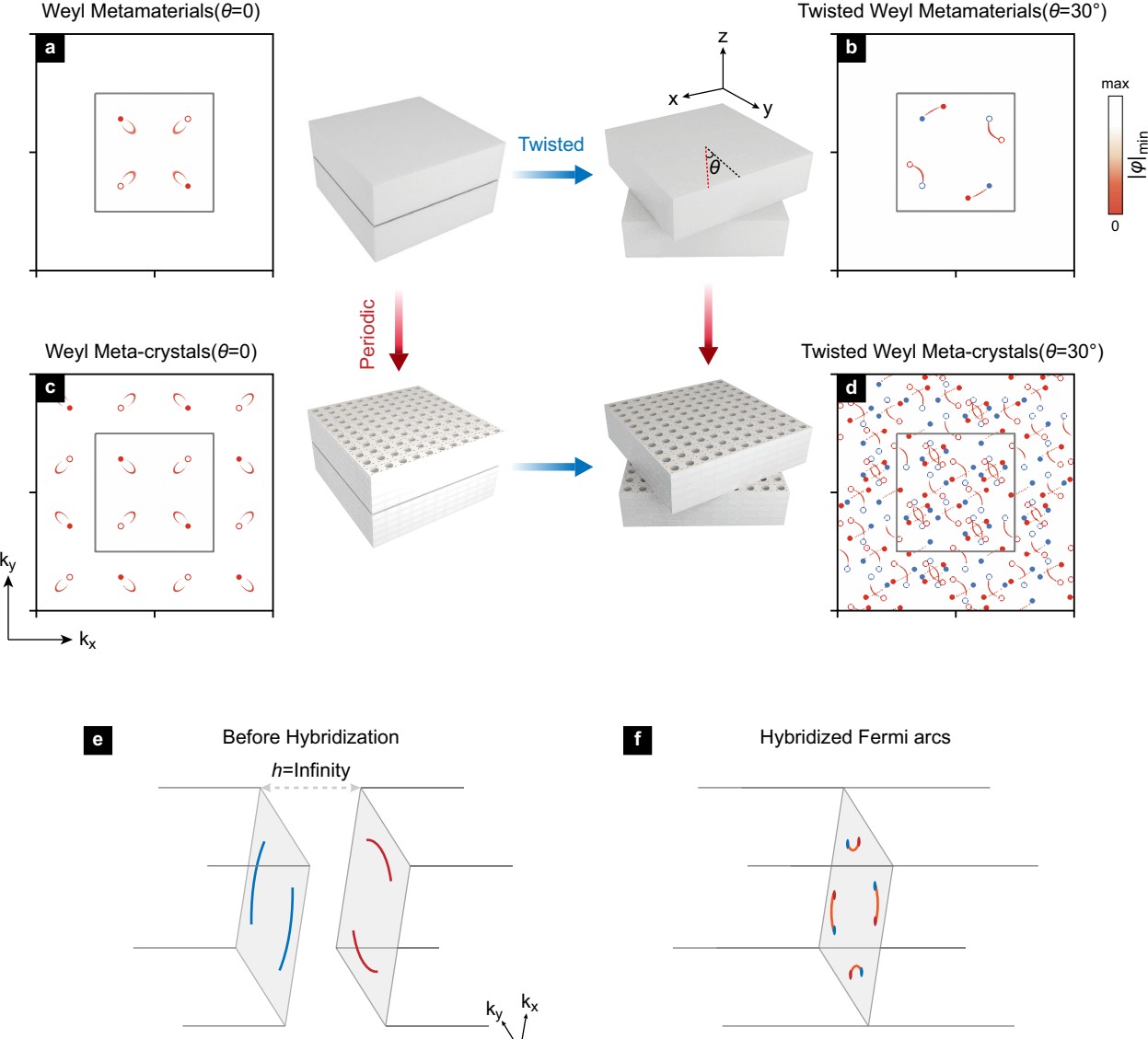

Fig. 1 | Dense Weyl Fermi arcs in a twisted bi-block meta-crystal. a Arcless surface states of Weyl metamaterials on a slightly gapped interface. b Fermi arcs of twisted bi-block Weyl metamaterials on the interface. There are eight Weyl points in total. The blue (red) color indicates the top (bottom) Weyl points, while circles and dots represent opposite charges. c Periodic spatial distribution of Weyl metamaterials constructs Weyl meta-crystal, where the arcless surface states are also periodically distributed. The gray square indicates the top Weyl meta-crystal's first Brillouin zone (top FBZ). d Dense distribution of twisted Weyl points and Fermi arcs on the interface of twisted bi-block Weyl meta-crystal. The corresponding real space schematic views are given accordingly. e Unhybridized Fermi arcs when the two surfaces are fully decoupled, e.g., $h \rightarrow \infty$. f Hybridized Fermi arcs when $h \rightarrow 0$ with the twisting angle $\theta = 30°$.

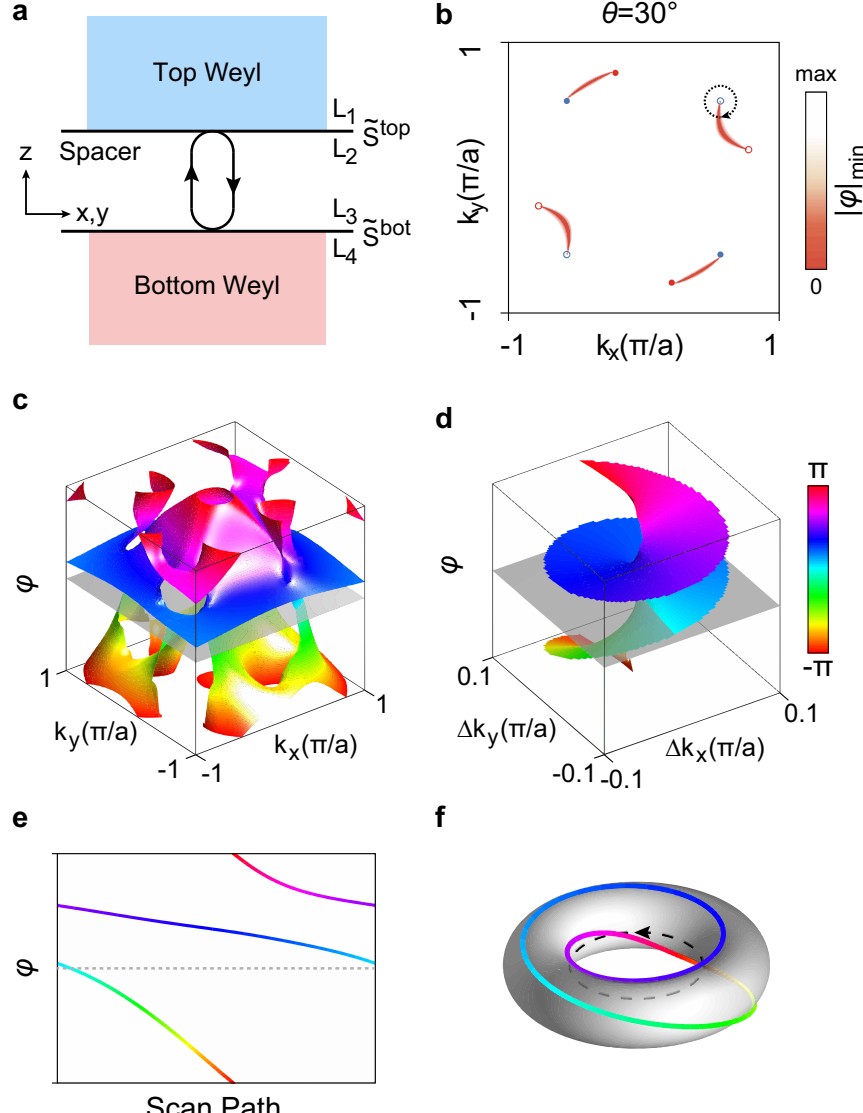

**Fig. 2 | Twisted bi-block Weyl metamaterials: Round-trip eigen phases and twisted Fermi arcs. a** Round-trip eigen phases and planar waveguide consisting of the top/bottom Weyl metamaterial and the middle spacer layer. **b** Interface Fermi arcs when $\theta = 30°$. **c** The round-trip eigen phase exhibits double-sheet Riemann surface structure. **d** The zoom-in view of the double-sheet Riemann surface structure in (**c**) around one Weyl point as encircled in (**b**). **e** The round-trip eigen phases around one Weyl point as indicated by the dashed circle in (**b**), which also corresponds to the helical boundary in (**d**). The horizontal dashed line denotes the guided mode condition $\varphi = \varphi^I + \varphi^{II} = 2n\pi$. **f** Torus knots built by mapping the round-trip eigen phases onto a torus.

twisting angle. However, it is not experimentally realistic. In other words, the realistic structure is periodically distributed in the spatial space. We have to apply meta-crystals to describe the experimental results.

**Twisted bi-block metamaterial**

For Weyl metamaterials, twisted Fermi arcs on the interface can be predicted via the round-trip eigen phase (see section II in Supplementary Information and Supplementary Fig. 2). As schematically shown in Fig. 2a, the twisted bi-block Weyl meta-crystal consists of three parts: top Weyl meta-crystal, dielectric spacer layer and bottom Weyl meta-crystal. The modes that can propagate in the spacer layer are the planar guided modes satisfying the condition $\varphi = \varphi^I + \varphi^{II} = 2n\pi$, where the topologically nontrivial phase $\varphi^I$ provided by reflection of Weyl meta-crystals, and trivial smooth phase $\varphi^{II}$ given by propagating in spacer layer. Specifically, the guided mode is described by the product of two scattering matrices $D = \widetilde{S}_{22}^{top}(\mathbf{k})\widetilde{S}_{11}^{bot}(\mathbf{k})$ (see Fig. 2a), which is a $2 \times 2$ unitary matrix with each eigenvalue taking the form of

$e^{i\varphi}$ that are the round-trip eigen phases. When we gradually decrease the height of spacer layer ($\varphi^{II} \to 0$), the planar guided modes finally turn to be interface modes, that are Fermi arc states (see Fig. 2b). And thus, we apply the planar guided mode condition to predict the interface Fermi arcs. Especially, when we focus on the Fermi arcs at Weyl frequency, both the top and bottom Weyl meta-crystals exhibit total reflection. Around each Weyl point, the round-trip eigen phases $\varphi$ are topologically non-trivial like the double-sheet Riemann surface (see Fig. 2c), where Weyl points serve as singularities[19]. In Fig. 2d we show a zoom-in view of the double-sheet Riemann surface, where the gray plane indicates the guided mode condition (interface Fermi arcs). Here, "double-sheet" is due to the two polarization states (e.g., TE and TM) of electromagnetic waves. The guided mode condition $\varphi = 2n\pi$ is also indicated by the dashed horizontal lines in Fig. 2e, which shows the round-trip eigen phases along the encircled loop in Fig. 2b. In Fig. 2e, for an arbitrary horizontal cut of the round-trip eigen phases, i.e., considering any extra trivial phase $\varphi^{II}$, there must be one point being crossed, which indicates that there must be one Fermi arc.

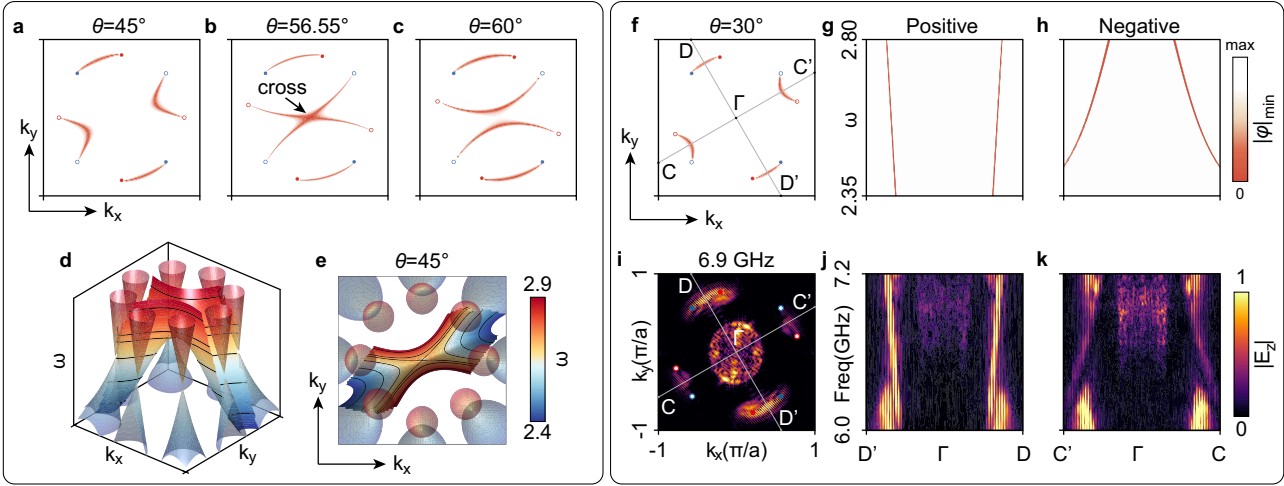

**Fig. 3 | Twisted bi-block Weyl metamaterials: Properties of twisted Fermi arcs.** **a**–**c** Lifshitz transition happens when continuously twisting the bi-block Weyl metamaterials. **d**, **e** Saddle-shaped surface band dispersion with the twisting angle $\theta = 45°$, which also corresponds to Lifshitz transition at the saddle point. **f**–**h** Positive and negative dispersions of twisted Fermi arcs. **i**–**k** The corresponding experimental results of (**f**–**h**).

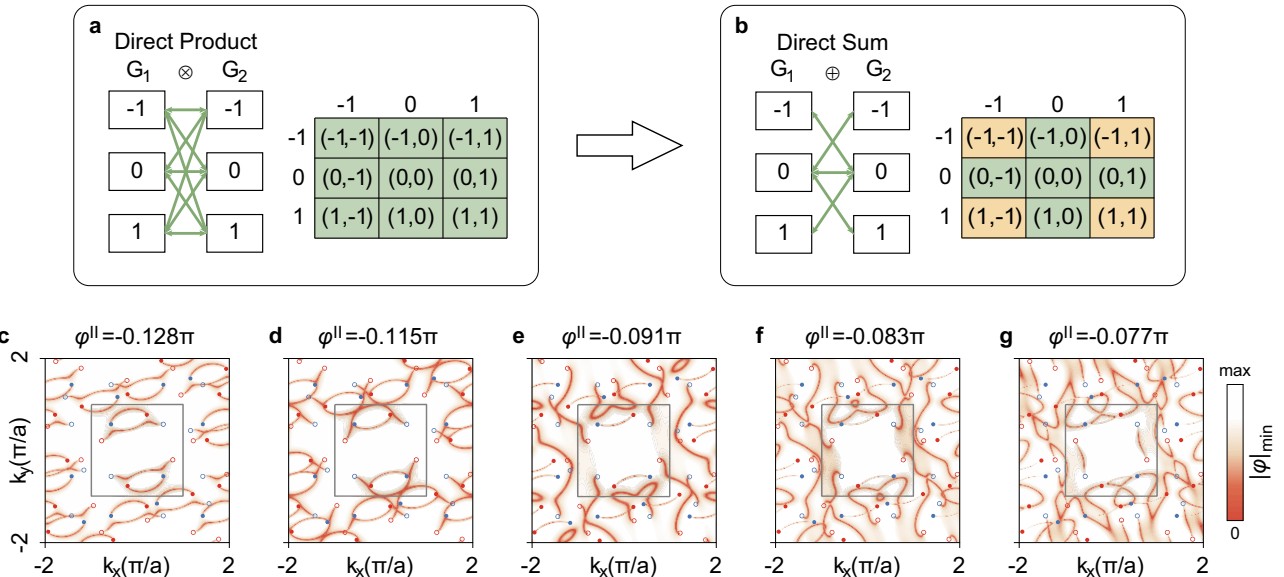

**Fig. 4 | Twisted bi-block Weyl meta-crystals: Twisted Fermi arc reconstruction.** **a**, **b** Direct Product and Direct Sum basis of the extended RCWA (rigorous coupled wave analysis) method, respectively. $G_1$ and $G_2$ are the collections of the basis of the top and bottom Weyl meta-crystals, respectively. Direct Product counts all possible combinations, while Direct Sum picks some as the green arrows indicated schematically. The colored green/yellow regions indicate picked/unpicked basis. **c**–**g** Reconstruction of twisted Fermi arcs when $\theta = 60°$ with the corresponding $\varphi^{II}$. The Fermi arcs reconnect other Weyl points accordingly.

Similarly, any mechanism that individually changes $\varphi^I$ or $\varphi^{II}$ leads to the reconstruction of Fermi arcs (see section III in Supplementary Information and Supplementary Figs. 3 and 4). In our work, the twisting angle $\theta$ reshapes $\varphi^I$ and thus enables continuous twisted Fermi arcs. The round-trip eigen phases actually exhibit the topology characteristic of Weyl points, we further map it onto a torus knot as shown in Fig. 2f.

For the twisted bi-block Weyl metamaterials, there are two pairs of twisted Fermi arcs as given in Fig. 3a, exhibiting convex and concave features. The concave twisted Fermi arcs show Lifshitz transition with increasing twisting angle $\theta$ (see Fig. 3a–c), where the concave Fermi arcs form a crossing point. The twisting angle is not the only parameter that leads to Lifshitz transition. Figure 3d, e show the Lifshitz transition when sweeping frequency. The band structure of concave Fermi arcs exhibits a saddle shape there. We also notice that the

concave Fermi arcs show distinct negative dispersion as shown in Fig. 3f–h. Scanning the surface states along two different paths (see Fig. 3f), the dispersion bands exhibit negative (positive) dispersion along $C − \Gamma − C'(D − \Gamma − D')$. Figure 3i–k show the corresponding experiment results (see below for experimental setups), respectively.

## Twisted bi-block meta-crystal

When the twisted angle is comparably large and the Fermi arcs are long enough, the scattering between quasi-periods tends to dominate the Fermi arc reconstruction. We have to consider the weak periodic modulation, that is Weyl meta-crystals rather than Weyl metamaterials, i.e., periodic modulation cannot be neglected any more. By introducing the RCWA (rigorous coupled wave analysis) method, which is used to study the scattering properties of photonic crystal slabs[49–52], we resolve the complex Fermi arc pattern in momentum space[49] (see

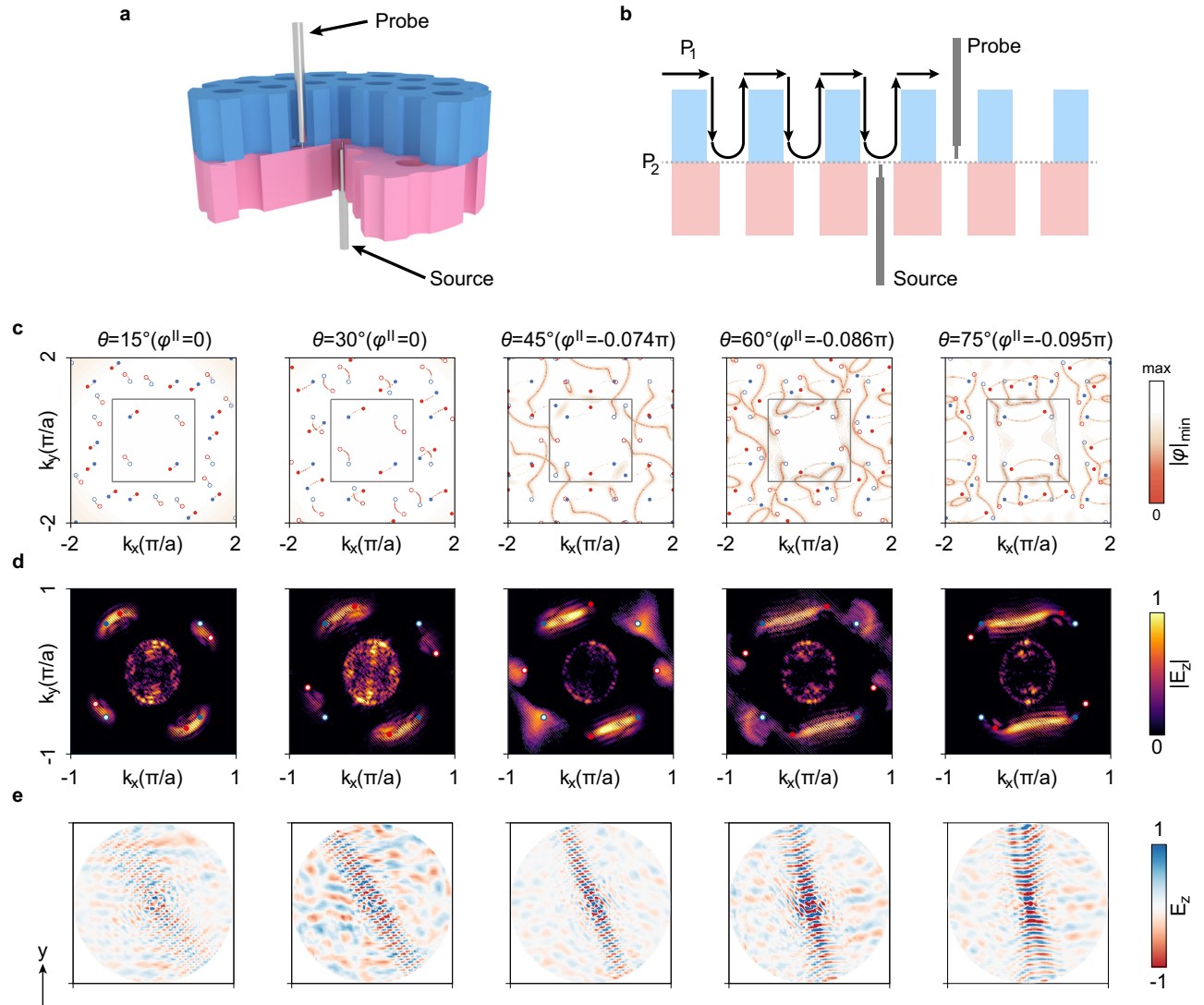

**Fig. 5 | Twisted bi-block Weyl meta-crystals: Experimental mapping of continuously twisted Fermi arcs. a** Schematic experimental setup of the twisted bi-block Weyl meta-crystal. The sample has been truncated into a cylinder shape for the continuous twisting purpose. Two portions have been cut to show the source and probe positions. The source antenna is inserted through the central hole of the bottom Weyl meta-crystal. **b** Schematic cutting view of the probe scanning path, where the probe antenna raster-scans the interface following a cycle manner ($P_1 \rightarrow P_2 \rightarrow P_1 \ldots$). **c** Fermi arcs predicted from the RCWA (rigorous coupled wave analysis) method. $\varphi^{II}$ stands for constant trivial phase, where we choose different values to fit the experimental results. **d** Experimentally mapped twisted Fermi arcs. **e** Continuously twisting of the topological interface waves that exhibit strong in-plane anisotropic propagation.

section IV in Supplementary Information). Different from considering the messy hoppings in real space as the usual tight-binding method, RCWA is very helpful in solving incommensurate twisting. Here, we extend the method to solve twisted Fermi arcs in twisted bi-block Weyl meta-crystals.

For the bi-block Weyl meta-crystals, the in-plane wave vectors **k** are extended into $\mathbf{k} + \mathbf{g}^{(1)} + \mathbf{g}^{(2)}$, where $\mathbf{g}^{(i)} \in G_i$ with $i = 1$ and 2 indicating the top and bottom blocks, respectively, and $G_i$ is the collection of the basis of the $i$ th block. Compared with the above Weyl meta-materials scattering matrix $\widetilde{S}^{\text{top/bot}}(\mathbf{k})$, here the scattering matrix $\widetilde{S}^i(\mathbf{k} + \mathbf{g}^{(1)} + \mathbf{g}^{(2)})$ depend on the plane wave basis $\mathbf{g}^{(1)} + \mathbf{g}^{(2)}$. Fermi arcs still could be predicted via the round-trip eigen phases $e^{i\varphi}$ similar to the above process (see Fig. 2a). The extended scattering matrix $\widetilde{S}^i(\mathbf{k} + \mathbf{g}^{(1)} + \mathbf{g}^{(2)})$ has dimensions of $4N^2 \times 4N^2$, with $N$ indicating the number of plane waves (NPW) in each block. Figure 4a schematically shows the Direct Product basis $G_1 \otimes G_2$. Theoretically, the more plane

waves are considered, the more Weyl points are involved. Leaving aside exponentially computational cost, one will obtain messy Weyl points and Fermi arcs bridging them (especially for incommensurate twisting angles). However, scattering energy is not uniformly distributed for all orders of Fermi arcs. Usually, the lower orders carry more scattering energy just like optical gratings, while other orders are relatively weak leading to only several twisted Fermi arcs bright.

We found that the Direct Sum of the plane wave basis could describe the dominant scattering nicely. As shown in Fig. 4b, the dimension of the scattering matrix is then reduced to $4(2N - 1) \times 4(2N - 1)$, where only specific components from the larger Direct Product scattering matrix are considered. To clearly show the dominating twisted Fermi arcs (aiming to fit the experimental results), we employ 9 plane waves (assuming a weak periodic modulation) for each Weyl meta-crystal (see section V in Supplementary Information and Supplementary Fig. 5). Figure 4c–g show the reconstruction of twisted Fermi arcs with changing $\varphi^{II}$, where twisted Fermi arcs go across the top FBZ boundary and connect to other Weyl points in

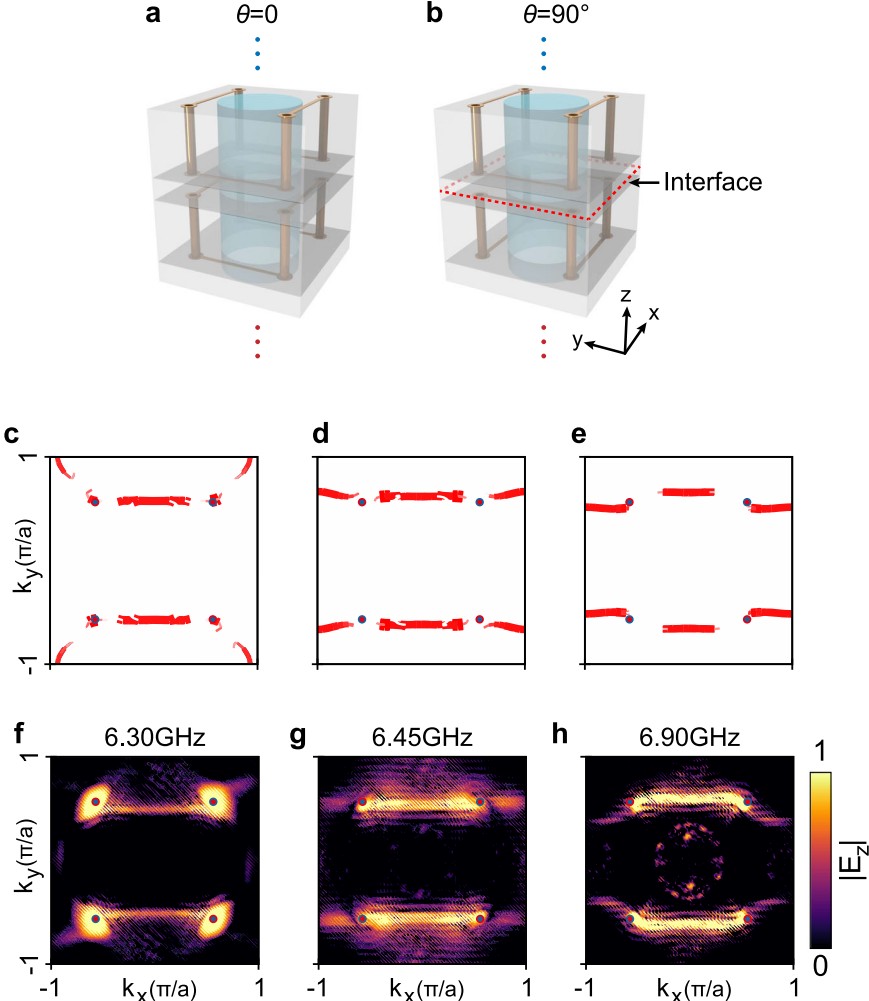

**Fig. 6 | Fermi arcs with $\theta$ = 90° preserving translation symmetry. a**, **b** The structure orientation closing to the interface. **c–e** Numerically simulated Fermi arcs. **f–h** Experimentally mapped Fermi arcs corresponding to (**c–e**).

succession. The twisting angle $\theta = 60°$ is an incommensurate angle, and so are the twisted Fermi arcs.

## Experimental results

To experimentally characterize the continuously twisted Fermi arcs, a near-field scanning system is employed, as schematically shown in Fig. 5a, b (see photography of the experimental setups in Supplementary Fig. 6 and section VI in Supplementary Information). The probe antenna can raster-scan the interface following the up and down cycles as shown in Fig. 5b. Figure 5c shows the theoretical results with twisting angles ranging from 15 to 75 degrees. One sees that the orientation of the twisted Fermi arcs can be continuously tuned. When $\theta \leq 30°$, both pairs of twisted Fermi arcs can be clearly observed, as predicted by Weyl metamaterials (see Supplementary Fig. 1). When $\theta > 30°$, one pair runs across the top FBZ boundary since the quasi-periodicity tends to dominate. We show the experimentally mapped twisted Fermi arcs at different twisting angles $\theta$ in Fig. 5d, which fit well with the theoretical predictions (see more angles in Supplementary Fig. 7 and section VII in Supplementary Information). The two pairs of twisted Fermi arcs are also revealed correspondingly in the real space field distribution, as shown in Fig. 5e. As twisting angle $\theta$ increases gradually, the electric field pattern rotates accordingly, exhibiting highly anisotropic features. The propagation pattern is also divided into two pairs, with the bright one corresponding to the positive dispersion (i.e., the convex pair), while the negative dispersion is comparably dimmer. Moreover, the Fermi arcs arising from scattering are

also observed, confirming the RCWA predictions (see section VIII in Supplementary Information and Supplementary Fig. 8).

## Discussion

Furthermore, we study the commensurate case when $\theta = 90°$ that preserves translation symmetry (see Fig. 6a, b). The periodic condition re-survives on the interface, and Fermi arcs can be studied as usual periodic systems as shown in Fig. 6c–e. The top and bottom Weyl points overlap and each projected Weyl point terminates two Fermi arcs[17]. The Fermi arcs emerge as two pairs, one bridges directly between Weyl points in the top FBZ while the other runs across the top FBZ boundary, as shown in Fig. 6c–e. The corresponding experimental results are given in Fig. 6f–h, which fit well with the simulation results (Fig. 6c–e).

The continuously tunable Fermi arcs in twisted bi-block Weyl meta-crystal offer new opportunities to explore a wide variety of possibilities. Firstly, the majority of natural systems lacks translational symmetry, i.e., quasicrystals and even disorder. The study of incommensurate lattices (broken translation invariance) could generalize those physics that were only believed to be stable under translation invariance. Weyl points and Fermi arcs belong to such systems and are both defined under translation symmetry, the generalization is fundamentally important. Here, we apply round-trip eigen phases to study interface Fermi arcs and show their incommensurate distribution in the momentum space. The conclusion works for all other systems, including electronic, photonic, acoustics, mechanics, etc. Secondly,

systems with broken translation symmetry have unique electronic, optical and mechanical properties[53,54]. In Weyl photonics, the spatial arrangement of Weyl points in incommensurate lattices could help in manipulating interfacial electromagnetic waves so as to continuously twist their propagation directions. Especially, the equi-frequency contour takes the form of arcs, which means strong intrinsic anisotropy[55], making them highly appealing for manipulating novel transmission or sensing phenomena. The scattering-induced twisted Fermi arcs, bound to the aperiodic interface, are also analogous to most possible aperiodic systems, such as the dense Fermi Bragg arcs in quasicrystal[47]. The twisted Fermi arcs in an aperiodic system also provide an outstanding scenario to examine properties in systems with incommensurate patterns but possessing long-range order[34]. In addition, the proposed scheme introduces arbitrarily controllable platforms for twisting Weyl physics as directly imposing a significant twist in a three-dimensional crystal is very challenging, e.g., the solid nature. Overall, the findings in twisted bi-block Weyl meta-crystals contribute substantially to understanding topological phenomena in condensed matter physics and may have implications for developing novel devices based on these aperiodic systems.

## Methods

### Sample fabrication

The sample was fabricated by a commercial company named Shenzhen Sunsoar Circuit Technology (http://www.oem-pcb.com) in Shenzhen, China, following the traditional Printed Circuit Board (PCB) fabrication technology. The technological details include: FR4 Material, board thickness 3.0 mm, copper weight 1 oz, no solder mask and surface leading OSP (Organic Solder-anility Preservative).

### Source and probe antennas

We employ a microwave vector network analyzer (VNA) and two near-field antennas acting as the source (stationary) and the probe (controlled by an *xyz* stage). The hole design limits the probe antenna's scan-step (which determines the maximum range of momentum space $\left[-\frac{\pi}{a}, \frac{\pi}{a}\right]$ after Fourier transformation) and *z* orientation (the probe is mostly sensitive to the field component that is parallel to its orientation). The antenna consists of a coaxial cable with a length of outer conductor and sheath stripped away, leaving the central conductor exposed, which can provide efficient coupling to large-momentum bulk and surface modes. The amplitude and phase of the near field are collected by the antenna with sub-wavelength resolution.

## Data availability

All data used in the analysis is available in Figshare with the identifier "https://doi.org/10.6084/m9.figshare.25288012".

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

## Acknowledgements

The authors sincerely thank Prof. Shuang Zhang from HKU for the fruitful discussion and support. This work is supported by the NSFC with Grant No. 12322412, NSFC with Grant No. 62275061, and National Key Research and Development Program of China (Grant No. 2022YFF0604802).

## Author contributions

B.Y., Q.G. and J.S. conceived the idea; H.W., B.Y., Q.G. and Z.Z. proposed the fabrication scheme; H.W. and Q.G. carried out all measurements; H.W., W.X., Z. Wei., Y.W., Q.G. and B.Y. carried out all simulations; H.W., W.X., Z. Wang., X.C., Q.G., J.S., Z.Z. and B.Y. developed and carried out the theoretical analysis; and Q.G., J.S., Z.Z. and B.Y. supervised the whole project. H.W., Q.G. and B.Y. wrote the paper and the Supplementary Information with input from all other authors.

## Competing interests

The authors declare no competing interests.
