## [Peer Review File · Nature Communications]

Twisted photonic Weyl meta-crystals and aperiodic Fermi arc scatteringReviewer #1 (Remarks to the Author):

In the manuscript by Wang et al. are analyzed the reconstruction of the Fermi arcs taking place at the interface between two Weyl semimetal photonic crystals (WMPCs). A Fermi arc is a chiral mode propagating unidirectionally on the surface of a WSPC and having an arc-like Fermi surface. They are topologically protected against disorder and defects and present intriguing phenomena such as Weyl orbits.

Generally, a WMPC is a three-dimensional photonic crystal hosting one or more pairs of Weyl points with opposite topological charges. The WMPC introduced in Ref. [9] and used as a basis system in the present manuscript is characterized by the presence of two pairs of Weyl points presenting evidence of Fermi arcs on opposite facets of the photonic crystal system.

In the present manuscript, the authors consider two of these WMPCs separated by an isotropic spacer layer of thickness h . The interface between these two WMPCs can result in both a commensurate or incommensurate configuration, depending on the relative rotation or twist angle. In the first case, it is possible to describe the interface between the two crystals as a crystal itself, whereas it is impossible in the second case due to the lack of translational symmetry. The authors present both a theoretical analysis and an experimental one. The theoretical one is based on the rigorous coupled wave analysis method presented in Ref. [39] that allows for solving photonic bilayer problems both in the commensurate and incommensurate configuration. The experiment is based on a type of WMPC crystal presented in Ref. [9] and authored by the last author of this manuscript. When the twist angle is zero, no reconstruction of the Fermi arcs is observed. However, when a finite twist is induced between the two WSPCs, the Fermi arc reconstruction occurs. In the new hybrid system, a Fermi arc is now formed between two Weyl points with the same topological charge but in different WMPC surfaces.

The paper presents for the first time the experimental evidence of the reconstruction of the Fermi arcs taking place at the interface between two photonic Weyl Semimetal systems. Previous experimental reports were for Fermi loops, which are Fermi arcs extending over all the first Brillouin zone [Liu et al., Nature 609, 925 (2022)]. In my opinion, the manuscript deserves publication in Nature Communication after some points have been addressed by the authors:

- 1) They should mention the manuscript about Fermi loop reconstruction in WSPCs: Liu et al., Nature 609, 925 (2022).
- 2) Extend the literature on the topic, for example, the work by Buccheri et al. [Phys. Rev. B 106, 045413 (2022)] about possible transport properties, Kaushik et al. [arXiv:2207.14109] elaborating on the magnetic oscillation; and Nguyen et al. [Phys. Rev. Lett. 131, 053602 (2023)] reporting on a possible observation of Fermi arcs reconstruction in a photonic lattice with synthetic momenta.
- 3) I found the abstract and the introduction of the manuscript misleading. The authors address a lot of the problem of topological physics in aperiodic systems, similar to the Ref. [20]. Here, any region inside the 3D Weyl semimetal quasicrystal has an aperiodic structure. However, in the present work, the only place where some aperiodic structure is found is at the interface between the two WSPCs. Each one of these is a perfect three-dimensional photonic crystal with four Weyl points and Fermi arcs on opposite surfaces (Ref. [9] by part of the authors). In this respect, this manuscript presents the first observation of Fermi arc reconstruction in a photonic system; all the rest is speculation.
- 4) The details regarding the photonic crystal are not mentioned in the main text. Even though the authors have cited Ref. [9] and the unit cell is shown in Section VII of the Supplementary Information, it is better to state clearly in the main text that the system of Ref. [9] is used here. Besides, relevant information such as the slab thickness or the spacer thickness compared to the lattice constant is not mentioned.
- 5) Figure 4(b) is misleading, the holes where the source antenna is placed, should be present on the bottom surface. These holes represent the one in the photonic crystal structure presented in Ref. [9]. If not, it should be clarified their origin and possible effect of commensurability with the photonic crystal.
- 6) The authors must define all the quantities introduced in the manuscript, for example, the vectors G at line 167.

7) Similarly, in II of the Supplemental Material, the constant c is used for the speed of light (set to one) and for another constant at line 66, set to one as well. Are they referring to the same quantity or is it a name duplication?

8) It is misleading that angles are defined using both radians and degrees; a uniform notation is appreciated by the reader of Nature Communications.

9) Regarding the round-trip eigenphase, the authors stress that the Fermi arcs are constructed after satisfying the round-trip eigenphase condition (line 122 and the SI); what is the specific relation between these two objects? When the spacer thickness is very large, do the interface Fermi arcs disappear, and is the condition $\phi + \phi' = 2n\pi$ still satisfied?

10) In Figure 2c, the authors state that "Torus knots are given to show the underlying topological features". What are the topological features referred to here?

11) In the relation $\phi + \phi' = 2n\pi$, ϕ' is a constant depending on the spacer thickness while ϕ varies continuously (Fig. 2b). How can this relation be satisfied?

Reviewer #2 (Remarks to the Author):

In this manuscript, the authors experimentally study Fermi-arc reconstruction at the interface between two photonic Weyl systems subject to a relative twist. They realize various different phases of this phenomenon, depending on twist angle, and show a complex landscape of Fermi-arcs, notably also under incommensurate twists.

The topic explored here is contemporary and would be of interest to a broad community of researchers. Furthermore, I find the results in the quasiperiodic case to be the most interesting as Quasicrystalline Weyl points were only recently predicted in Phys. Rev. B 108, L121109 (Ref. 20). The manuscript could therefore be suitable for publication in Nature Communications. However, the presentation and writing is not up to the mark and requires substantial revision in my opinion (some points are detailed below). I also have a few questions for the authors to consider:

1. How would the Fermi-arc connectivity would change if the two Weyl semimetals were inverted with respect to each other, i.e., in the case where the charges of the Weyl points in one block are opposite in sign to their counterparts in the other block. Is there a reason such a system wasn't chosen for exploration?

2. Due to the relatively small z-extent of the blocks, are there any finite size effects that emerge from the top and bottom surfaces (the surfaces with vacuum/air)? If the system is made even smaller along z, could the Fermi-arc states at the interface be resolved using transmission measurements through the full bi-block system?

Manuscript issues:

1. In Fig. 1, typical values of θ could be mentioned in the (a) - (e). In the same figure, the real space dimensions of the bi-block patterning can be provided. Another subfigure that could be included to clearly show the Fermi-arcs for each block and then their reconstruction. From the present figure, it is hard to understand how the Fermi-arcs hybridize.

2. Maybe I missed this but I found no mention about the operating frequency, except for a small label in Fig. 2(m); this is important to mention in the main text. Similarly, the typical dimensions of the bi-block, material, design of the unit cell and lattice constant etc. also cannot be found. A methods section seems to be missing from the supplementary with enough detail to reproduce the experiment. Fig. 4 (a) and (b) do not provide a clear understanding of the experimental setup and these can be also improved substantially.

3. The writing is very clunky and parts of the manuscript are hard to understand. This especially applies to the round-trip eigenphase discussion and the RCWA paragraph. A more pedagogical

approach here would allow for a wider audience to understand these key points.

Reviewer #3 (Remarks to the Author):

The work is about an artificial photonic Weyl meta-crystal featuring an interface between two blocks of photonic Weyl crystals that can be continuously rotated with respect to each other. The authors focus on the topological interface Fermi arcs, predicted to occur at such interfaces. The Fermi arcs are experimentally found via exciting and raster-scanning the photonic modes at the interface. The experimental findings are compared with extensive theoretical calculations of the Fermi arcs via a coupled-wave analysis. The work constitutes a first experimental evidence of interface Fermi arcs and, besides their bare presence, the authors reveal and analyse the Fermi arc reconstruction due to interaction with an aperiodic lattice.

The findings of this work have a high level of significance. The overall analysis, experimental data, and presentation of the results seem sound. I have however a concern, whether enough evidence is given that the experimentally observed modes really are Fermi arcs. If the authors can clarify the role of the topological round-trip eigenphase (see comments below), that might resolve this problem.

If the authors can address this and the other points listed below satisfactorily, I would support the publication of the manuscript.

Here the detailed comments and questions, labeled by the line numbers of the manuscript:

34: "Singular reflection phase" is unclear at that point.

41-43: Clarify that translation invariance in the plane parallel to the interface is meant. (If I understand it correctly).

45-46: What is the significance of broken translation invariance with regard to applications and what are such applications?

51: The phrase "interaction between topology and physics" is unclear.

95-97: The terminology Weyl meta-materials and meta-crystals needs to be explained better at this point. I understood what is meant only after reading the sentences in lines 155-158.

111: What exactly is the round-trip eigenphase? How exactly are the round-trip eigenphases used here? Are they useful to identify the Fermi arcs theoretically or are they used to identify the Fermi arcs experimentally?

118: In Fig 2b,c it looks like a phase of 2π is accumulated by going around the Weyl node *twice*. This appears contradictory to what is stated in the text.

205: What are the "up and down cycles" and why are they necessary for the scanning? Fig 4b seems to be not very useful in the description.

227: Fig. 4 a and b caption is too short. At least a hint could be given that the setup is described in the text.

228: With regard to the strong dependence of the arcs on the fitting parameter ϕ' , shown in Fig.3, please explain the value of the experiment - theory correspondence.

RESPONSE TO REVIEWERS

Reviewer #1 (Remarks to the Author):

Comment 1.1: In the manuscript by Wang et al. are analyzed the reconstruction of the Fermi arcs taking place at the interface between two Weyl semimetal photonic crystals (WMPCs). A Fermi arc is a chiral mode propagating unidirectionally on the surface of a WSPC and having an arc-like Fermi surface. They are topologically protected against disorder and defects and present intriguing phenomena such as Weyl orbits.

Generally, a WMPC is a three-dimensional photonic crystal hosting one or more pairs of Weyl points with opposite topological charges. The WMPC introduced in Ref. [9] and used as a basis system in the present manuscript is characterized by the presence of two pairs of Weyl points presenting evidence of Fermi arcs on opposite facets of the photonic crystal system.

In the present manuscript, the authors consider two of these WMPCs separated by an isotropic spacer layer of thickness h . The interface between these two WMPCs can result in both a commensurate or incommensurate configuration, depending on the relative rotation or twist angle. In the first case, it is possible to describe the interface between the two crystals as a crystal itself, whereas it is impossible in the second case due to the lack of translational symmetry. The authors present both a theoretical analysis and an experimental one. The theoretical one is based on the rigorous coupled wave analysis method presented in Ref. [39] that allows for solving photonic bilayer problems both in the commensurate and incommensurate configuration. The experiment is based on a type of WMPC crystal presented in Ref. [9] and authored by the last author of this manuscript. When the twist angle is zero, no reconstruction of the Fermi arcs is observed. However, when a finite twist is induced between the two WSPCs, the Fermi arc reconstruction occurs. In the

new hybrid system, a Fermi arc is now formed between two Weyl points with the same topological charge but in different WMPC surfaces.

Reply 1.1: We thank the referee for the careful reading and the nice summary of our work.

Comment 1.2: The paper presents for the first time the experimental evidence of the reconstruction of the Fermi arcs taking place at the interface between two photonic Weyl Semimetal systems. Previous experimental reports were for Fermi loops, which are Fermi arcs extending over all the first Brillouin zone [Liu et al., Nature 609, 925 (2022)]. In my opinion, the manuscript deserves publication in Nature Communication after some points have been addressed by the authors:

Reply 1.2: We thank the referee for his/her recognitions and positive assessments of our work. We also thank the referee for the following comments to improve the presentation of our work.

Comment 1.3: 1) They should mention the manuscript about Fermi loop reconstruction in WSPCs: Liu et al., Nature 609, 925 (2022).

Reply 1.3: We thank the referee for bringing to our attention the interesting work.

In the revised main text, we have added the following text in the introduction and cited the work as,

“... just like the reconstruction of Fermi loop surface states, which have been cleverly demonstrated in magnetically tunable 3D photonic crystals¹⁷ with the closed topological surface states extending across the surface Brillouin zone in the forms of torus knots.”.

Comment 1.4: 2) Extent the literature on the topic, for example, the work by Buccheri et al. [Phys. Rev. B 106, 045413 (2022)] about possible transport properties, Kaushik et al. [arXiv:2207.14109] elaborating on the magnetic oscillation; and Nguyen et al. [Phys. Rev. Lett. 131, 053602 (2023)] reporting on a possible observation of Fermi arcs reconstruction in a photonic lattice with synthetic momenta.

Reply 1.4: We sincerely thank the referee for bringing to our notice these very related works. In the revised main text, we have added the following text and cited them in the introduction as,

“Very recently, manipulating Fermi arcs has further stimulated more interests, e.g. possible transport properties in junctions of Weyl semimetals¹¹, quantized chiral magnetic current¹², and Fermi arc reconstruction in synthetic photonic lattice¹³, etc. Understanding, constructing and manipulating Fermi arcs¹⁴⁻¹⁶ can lead to the development of novel electronic/photonic devices and quantum technologies, just like the reconstruction of Fermi loop surface states, which have been cleverly demonstrated in magnetically tunable 3D photonic crystals¹⁷ with the closed topological surface states extending across the surface Brillouin zone in the forms of torus knots.”.

Comment 1.5: 3) I found the abstract and the introduction of the manuscript misleading. The authors address a lot of the problem of topological physics in aperiodic systems, similar to the Ref. [20]. Here, any region inside the 3D Weyl semimetal quasicrystal has an aperiodic structure. However, in the present work, the only place where some aperiodic structure is found is at the

interface between the two WSPCs. Each one of these is a perfect three-dimensional photonic crystal with four Weyl points and Fermi arcs on opposite surfaces (Ref. [9] by part of the authors). In this respect, this manuscript presents the first observation of Fermi arc reconstruction in a photonic system; all the rest is speculation.

Reply 1.5: We apologize for the ambiguous and misleading parts in the abstract and introduction, and we thank the referee for pointing this out.

Indeed, in the twisted bi-block photonic Weyl meta-crystals, only the interface exhibits aperiodic features, while both the Weyl meta-crystals individually preserve translational symmetry.

In the revised main text, we have revised the statement in the abstract as,

“However, no experimental observation of aperiodic Fermi arcs has been reported so far. Here, we continuously twist a bi-block Weyl meta-crystal and experimentally observe the twisted Fermi arc reconstruction. Although both the Weyl meta-crystals individually preserve translational symmetry, continuous twisting operation leads to the aperiodic hybridization and scattering of Fermi arc on the interface, which is found to be determined by the singular total reflection around Weyl points.”.

And in the introduction, we have further emphasized as,

“It is worth mentioning that only the interface exhibits aperiodic features, while both the Weyl meta-crystals individually preserve translational symmetry.”.

Comment 1.6: 4) The details regarding the photonic crystal are not mentioned in the main text. Even though the authors have cited Ref. [9] and the unit cell is shown in Section VII of the Supplementary Materials, it is better to state clearly in the main text that the system of Ref. [9] is

used here. Besides, relevant information such as the slab thickness or the spacer thickness compared to the lattice constant is not mentioned.

Reply 1.6: We thank the referee for pointing this out.

In the revised Supplementary Materials, we have added Fig. S6a to show the detailed dimension parameters of the Weyl meta-crystal used. We have copied Fig. S6 (see Fig. R1) below for convenience.

Fig. R1 (Fig. S6). Details of the experimental sample. **a**, Dimension parameters of Weyl meta-crystal used in the experiment. The size of plated through holes depends on the specific printed circuit board (PCB) technology and will not influence the results. The copper thickness is 1oz

(35 μm). **b**, The top and bottom Weyl meta-crystals are relatively twisted by an angle θ . **c**, Top view of the sample. The central red point indicates the center hole, where the source antenna inserted from the bottom block. There are 71 unit-cells along both x and y directions (i.e. its diameter has 71 unit-cells). **d**, The zoom-in view. The probe antenna raster-scans the periodic holes (see details in Method).

Comment 1.7: 5) Figure 4(b) is misleading, the holes where the source antenna is placed, should be present on the bottom surface. These holes represent the one in the photonic crystal structure presented in Ref. [9]. If not, it should be clarified their origin and possible effect of commensurability with the photonic crystal.

Reply 1.7: We apologize for the misleading schematic plot in Fig. 4b.

The Weyl meta-crystal structure used here is slightly different from that in Ref. 9 [Science 359, 1013-1016 (2018)]. And the holes in Ref. 9 are the plated through holes in the printed circuit board (PCB) technology. In current work, we additionally drill a central hole (with the diameter of 3.8mm) in each unit cell (see Fig. R1a above) to make sure that the probe/source antenna can reach the interface as shown in Fig. R2 (Figs. 4a and b). In other words, these drilled holes are designed only for experimental purpose and thus preserve all the symmetries (i.e. the commensurability with the original photonic crystals).

In the revised main text, we have revised Figs. 4a and b (copied below as Fig. R2 for convenience).

Fig. R2 (Figs. 4a-b). **a**, Schematic experimental setup of the twisted bi-block Weyl meta-crystal. The sample has been truncated into a cylinder shape for the continuous twisting purpose. Two portions have been cut to show the source and probe positions. The source antenna is inserted through the central hole of the bottom Weyl meta-crystal. **b**, Schematic cutting view of the probe scanning path, where the probe antenna raster-scans the interface following a cycle manner ($P_1 \rightarrow P_2 \rightarrow P_1 \dots$).

Comment 1.8: 6) The authors must define all the quantities introduced in the manuscript, for example, the vectors G at line 167.

Reply 1.8: We thank the referee for pointing this out.

In the revised turn, we have thoroughly checked all the definitions of quantities. The vectors G are defined as,

“For the bi-block Weyl meta-crystals, the in-plane wave vectors \vec{k} are extended into $\vec{k} + \vec{g}^{(1)} + \vec{g}^{(2)}$, where $\vec{g}^{(i)} \in G_i$ with $i = 1$ and 2 indicating the top and bottom blocks, respectively, and G_i is the collection of the basis of the i th block.”.

Comment 1.9: 7) Similarly, in II of the Supplemental Material, the constant c is used for the speed of light (set to one) and for another constant at line 66, set to one as well. Are they referring to the same quantity or is it a name duplication?

Reply 1.9: We thank the referee for pointing this out. It is a name duplication. The second “ c ” refers to one of nonlocal coefficients (we have already replaced it with “ γ ”). In the revised turn, we have carefully revised all the similar duplication.

Comment 1.10: 8) It is misleading that angles are defined using both radians and degrees; a uniform notation is appreciated by the reader of Nature Communications.

Reply 1.10: We thank the referee for pointing this out. We have unified the use of angles and radians in the revised turn.

Comment 1.11: 9) Regarding the round-trip eigen phase, the authors stress that the Fermi arcs are constructed after satisfying the round-trip eigen phase condition (line 122 and the SI); what is the specific relation between these two objects? When the spacer thickness is very large, do the interface Fermi arcs disappear, and is the condition $\phi + \phi' = 2n\pi$ still satisfied?

Reply 1.11: We thank the referee for the nice question.

In brief, the topological interface states are the planar guided modes sandwiched between the two Weyl meta-crystals. Therefore, we use the round-trip eigen phase condition (or guided mode condition) $\varphi = \varphi^I + \varphi^{II} = 2n\pi$ to predict the interface Fermi arcs, where φ^I (φ^{II}) denotes the nontrivial (trivial) part of round-trip eigen phase.

As schematically shown in Fig. R3a, the twisted bi-block Weyl meta-crystal consists of three parts: top Weyl meta-crystal, dielectric spacer layer and bottom Weyl meta-crystal. The modes that can propagate in the spacer layer are the planar guided modes. When we gradually decrease the height of the spacer layer, the planar guided modes finally turn to be interface modes, that are Fermi arc states here. And thus, we apply the planar guided mode condition $\varphi = \varphi^I + \varphi^{II} = 2n\pi$ to predict the interface Fermi arcs. Especially, when we focus on the Fermi arcs at Weyl frequency, both the top and bottom Weyl meta-crystals exhibit total reflection. Around each Weyl point the total reflection phase is topologically nontrivial like Riemann surface (see Fig. R3b), i.e. providing the topological phase φ^I , which is the key difference from conventional waveguides.

In Fig. R3c, we show the twisted Fermi arcs directly calculated from boundary condition matching method, which are exactly the same as those predicted from the round-trip eigen phase condition as shown in Fig. R3d. Therefore, the round-trip eigen phase condition can well describe interface Fermi arcs.

When the spacer thickness is very large, the configuration is still a planar waveguide. And thus the guided mode condition $\varphi = 2n\pi$ remains unchanged. However, the trivial phase φ^{II} tends to dominate while the topological phase φ^I stays the same, so the interface Fermi arcs are hidden somehow as shown in Figs. R3e-g. It is very difficult to recognize the pattern of Fermi arcs.

In the revised main text, we have replotted Fig. 2 to elaborately explain the relation between the round-trip eigen phases and interface Fermi arcs.

Fig. R3. **a**, Configuration of planar waveguide, which consists of the top/bottom Weyl meta-crystal and the middle spacer layer. **b**, The round-trip eigen phase exhibits Riemann surface structure. **c**, Interface Fermi arcs calculated by directly matching boundary condition. **d**, Interface Fermi arcs predicted by the round-trip eigen phase condition (or guided mode condition) $\varphi = \varphi^I + \varphi^{II} = 2n\pi$. **e-g**. The planar guided modes for different spacer layer thickness h , whose dielectric constant $\epsilon = 16$. All the twisting angle $\theta = 30^\circ$.

Comment 1.12: 10) In Figure 2c, the authors state that “Torus knots are given to show the underlying topological features”. What are the topological features referred to here?

Reply 1.12: We thank the referee for the nice question.

When we scan around one Weyl point once (see the dashed circle in Fig. R4a), at each k point there are two (round-trip) eigen phases as shown in Fig. R4b. While for each φ (assuming a horizontal dashed line in Fig. R4b), there is only one crossing point, which implies one Fermi arc state. Furthermore, the eigen phases are periodic along both the scan path and φ axis. It could be

more obvious when we map the two eigen phases to be a single torus knot as shown in Fig. R4c, which intuitively exhibits the underlying topological features of the round-trip eigen phases. In summary, it refers to the topological features of Weyl point encircled.

In the revised main text, we have replotted Fig. 2 to elaborately explain the round-trip eigen phase.

Fig. R4. **a**, Fermi arcs when the twisting angle $\theta = 60^\circ$. **b**, Round-trip eigen phases around one Weyl point, see the scan path as dashed circle in (a) with the start and end points meeting at the arrow position. The horizontal dashed line denotes the guided mode condition $\varphi = \varphi^I + \varphi^{II} = 2n\pi$. **c**, Torus knot built by mapping the round-trip eigen phases on to a torus.

Comment 1.13: 11) In the relation $\varphi + \varphi' = 2n\pi$, φ' is a constant depending on the spacer thickness while φ varies continuously (Fig. 2b). How can this relation be satisfied?

Reply 1.13: We thank the referee for the nice question.

In Fig. 2b, we show the round-trip eigen phases φ varying from $-\pi \rightarrow \pi$ when the path encircled one Weyl point, there must be a single zero point that is $\varphi = 0$. Therefore, the relation is always satisfied when $\varphi^{II} = 0$ (In the revised turn, for brevity we have made the replacements $\varphi \rightarrow \varphi^I$ and $\varphi' \rightarrow \varphi^{II}$). When $\varphi^{II} \neq 0$ the relation could be still satisfied as long as φ^{II} is topologically trivial as shown in Fig. R5. Usually, φ^{II} is introduced by the trivial spacer layer, that means no singularity (unlike φ^I). We show several φ^{II} with changing the thickness as shown in Figs. R5a-d, one sees there is no any singularity. Accordingly, Fermi arcs varies as shown in Figs. R5e-g,

with the corresponding round trip eigen phases shown in Figs. R5h-j. In a word, different φ^{II} just rotates Fermi arc state around each Weyl point while the relation $\varphi = \varphi^I + \varphi^{II} = 2n\pi$ is always satisfied. In the revised Supplementary Materials, we have added above discussion in Section III and Fig. S4.

Fig. R5 (Fig. S4). Fermi arc changing with trivial phase φ^{II} . **a-d**, Different φ^{II} corresponding to different dielectric spacer layer thickness h with the dielectric constant $\epsilon = 16$. **e-g**, Fermi arcs change for different h , the twisting angle $\theta = 60^\circ$. **h-j**, Round-trip eigen phases around Weyl point, where the scan paths are indicated in (e-g) with the start and end points meeting at the arrow position. The horizontal dashed lines denote the guided mode condition $\varphi = \varphi^I + \varphi^{II} = 2n\pi$.

Reviewer #2 (Remarks to the Author):

Comment 2.1: In this manuscript, the authors experimentally study Fermi-arc reconstruction at the interface between two photonic Weyl systems subject to a relative twist. They realize various different phases of this phenomenon, depending on twist angle, and show a complex landscape of Fermi-arcs, notably also under incommensurate twists.

The topic explored here is contemporary and would be of interest to a broad community of researchers.

Reply 2.1: We thank the referee for the nice summary of our work.

Comment 2.2: Furthermore, I find the results in the quasiperiodic case to be the most interesting as Quasicrystalline Weyl points were only recently predicted in Phys. Rev. B 108, L121109 (Ref. 20). The manuscript could therefore be suitable for publication in Nature Communications.

Reply 2.2: We thank the referee for his/her recognitions of our work.

Comment 2.3: However, the presentation and writing is not up to the mark and requires substantial revision in my opinion (some points are detailed below).

Reply 2.3: We thank the referee for careful reading and the following comments to highly improve our presentation.

Comment 2.4: I also have a few questions for the authors to consider:

1. How would the Fermi-arc connectivity would change if the two Weyl semimetals were inverted with respect to each other, i.e., in the case where the charges of the Weyl points in one block are opposite in sign to their counterparts in the other block. Is there a reason such a system wasn't chosen for exploration?

Reply 2.4: We thank the referee for the nice question.

The case mentioned above actually corresponds to the twisting angle $\theta = 90^\circ$, which is indeed intriguing and warrants thorough exploration. As our main text focuses on the incommensurate twisting angle, previously we discussed it (commensurate ones) only in Supplementary Materials. In the revised turn, we have moved it into the main text and added Fig. 6 accordingly.

Given that the coordinates of the four Weyl points in our ideal Weyl meta-crystal are $(\pm WP, \pm WP)$, a 90-degree rotation of the bottom Weyl meta-crystal makes the inversion of chirality. The positive/negative Weyl points in the top Weyl meta-crystal precisely overlap with the negative/positive ones in the bottom. On the interface, each projected Weyl point carries the topological charge of 2 and thus connects to two Fermi arcs, as shown in Fig. R6c. The numerically simulated and the experimentally measured Fermi arcs are shown in Figs. R6c-e and Figs. R6f-h, respectively.

Fig. R6 (Fig. 6). Fermi arcs with $\theta = 90^\circ$ preserving translation symmetry. **a and b**, The structure orientation closing to the interface. **c-e**, Numerically simulated Fermi arcs. **f-h**, Experimentally mapped Fermi arcs corresponding to (c-e).

Comment 2.5: 2. Due to the relatively small z -extent of the blocks, are there any finite size effects that emerge from the top and bottom surfaces (the surfaces with vacuum/air)?

Reply 2.5: We thank the referee for the nice question.

In the experiment, we use 15 periods for each Weyl meta-crystal block in the z direction to eliminate the influence of the surface states on the top and bottom air surfaces.

In the twisted bi-block Weyl meta-crystal system, apart from the aperiodic Fermi arcs induced by the twist at the interface (which is our main point), there are also air surface states on the top and bottom surfaces. But these top/bottom surface states cannot propagate in Weyl meta-crystal, i.e.

exponentially decay along the z -direction. If the z -extent of each block is small enough, e.g. smaller than the decay length (about 5 periods), the top/bottom surface states will surely couple with the interface Fermi arcs and influence our results.

Comment 2.6: If the system is made even smaller along z , could the Fermi-arc states at the interface be resolved using transmission measurements through the full bi-block system?

Reply 2.6: We thank the referee for the enlighten question. In the revised turn, we have followed the referee's suggestion and scanned a much thinner bi-block twisted Weyl meta-crystal. As shown in Fig. R7, the interfaces states are still readable although much dimmer compared with those collected on the interface.

Fig. R7. Result comparison between interface scanning and transmission measurement with twisting angle $\theta = 30^\circ$ at 6.9 GHz. **a**, The probe antenna raster-scans the interface following a cycle manner ($P_1 \rightarrow P_2 \rightarrow P_1 \dots$). **b-c**, Fermi arcs in momentum space and the corresponding wave

propagation in the real space collected by the setup in (a). **d**, The probe antenna raster-scans the top surface. **e–f**, Fermi arcs and their real space distribution collected by the setup in (d), which are quite dim compared with those collected in the setup (a).

Comment 2.7: Manuscript issues:

1. In Fig. 1, typical values of theta could be mentioned in the (a) - (e). In the same figure, the real space dimensions of the bi-block patterning can be provided. Another subfigure that could be included to clearly show the Fermi-arcs for each block and then their reconstruction. From the present figure, it is hard to understand how the Fermi-arcs hybridize.

Reply 2.7: We thank the referee for improving our presentation in Fig. 1. In the revised turn, we have fully updated Fig. 1 as copied below (see Fig. R8) for convenience.

1. We added typical values of theta for each twisting case.
2. We provided the real space dimensions of the bi-block patterning.
3. We added two new panels to clearly show the (un)hybridized Fermi arcs.

Fig. R8 (Fig. 1). **Dense Fermi arcs in a twisted bi-block Weyl meta-crystal.** **a**, Arcless surface states of Weyl metamaterials on a slightly gapped interface. **b**, Fermi arcs of twisted Weyl metamaterials on the interface. There are eight Weyl points in total. The blue (red) color indicates the top (bottom) Weyl points, while circles and dots represent opposite charges. **c**, Periodic spatial distribution of Weyl metamaterials constructs Weyl meta-crystal, where the arcless surface states are also periodically distributed. The gray square indicates the top Weyl meta-crystal's first Brillouin zone (top FBZ). **d**, Dense distribution of twisted Weyl points and Fermi arcs on the interface of twisted bi-block Weyl meta-crystal. The corresponding real space schematic views are given accordingly. **e**, Unhybridized Fermi arcs when the two surfaces are fully decoupled, e.g. $h \rightarrow \infty$. **f**, Hybridized Fermi arcs when $h \rightarrow 0$ with the twisting angle $\theta = 30^\circ$.

Comment 2.8: 2. Maybe I missed this but I found no mention about the operating frequency, except for a small label in Fig. 2(m); this is important to mention in the main text.

Reply 2.8: We apologize for the missing information. In the revised turn, we have fully checked and added all the operating frequencies accordingly.

Comment 2.9: Similarly, the typical dimensions of the bi-block, material, design of the unit cell and lattice constant etc. also cannot be found.

Reply 2.9: We apologize for the missing information and thank for the referee's careful check. In the revised turn, we have added Fig. S6 (as copies below for convenience) in the Supplementary Materials to show geometric dimension details.

Fig. R9 (Fig. S6). Details of the experimental sample. **a**, Dimension parameters of Weyl meta-crystal used in the experiment. The size of plated through holes depends on the specific printed circuit board (PCB) technology and will not influence the results. The copper thickness is 1oz (35 μ m). **b**, The top and bottom Weyl meta-crystals are relatively twisted by an angle θ . **c**, Top view of the sample. The central red point indicates the center hole, where the source antenna inserted from the bottom block. There are 71 unit-cells along both x and y directions (i.e. its diameter has 71 unit-cells). **d**, The zoom-in view. The probe antenna raster-scans the periodic holes (see details in Method).

Comment 2.10: A methods section seems to be missing from the supplementary with enough detail to reproduce the experiment. Fig. 4 (a) and (b) do not provide a clear understanding of the experimental setup and these can be also improved substantially.

Reply 2.10: We thank the referee for pointing this out. In the revised turn, we have added a Materials and Methods section in Supplementary Materials to show the experiment details. In addition, we have updated Fig. 4 (copied below as Fig. R10 for convenience) to further schematically show the experiment setups.

Fig. R10 (Figs. 4a-b). **a**, Schematic experimental setup of the twisted bi-block Weyl meta-crystal. The sample has been truncated into a cylinder shape for the continuous twisting purpose. Two portions have been cut to show the source and probe positions. The source antenna is inserted through the central hole of the bottom Weyl meta-crystal. **b**, Schematic cutting view of the probe scanning path, where the probe antenna raster-scans the interface following a cycle manner ($P_1 \rightarrow P_2 \rightarrow P_1 \dots$).

Comment 2.11:3. The writing is very clunky and parts of the manuscript are hard to understand. This especially applies to the round-trip eigenphase discussion and the RCWA paragraph. A more pedagogical approach here would allow for a wider audience to understand these key points.

Reply 2.11: We thank the referee for highly improving our presentation. In the revised turn, we have re-written “the round-trip eigen phase discussion (see Fig. 2) and the RCWA paragraph” and thoroughly proof read all texts. We quite appreciate for the referee’s warm suggestions and hope that the revised texts are pedagogical enough for a wider audience.

Reviewer #3 (Remarks to the Author):

Comment 3.1: The work is about an artificial photonic Weyl meta-crystal featuring an interface between two blocks of photonic Weyl crystals that can be continuously rotated with respect to each other. The authors focus on the topological interface Fermi arcs, predicted to occur at such interfaces. The Fermi arcs are experimentally found via exciting and raster-scanning the photonic modes at the interface. The experimental findings are compared with extensive theoretical calculations of the Fermi arcs via a coupled-wave analysis.

Reply 3.1: We thank the referee for careful reading and the nice summary of our work.

Comment 3.2: The work constitutes a first experimental evidence of interface Fermi arcs and, besides their bare presence, the authors reveal and analyse the Fermi arc reconstruction due to interaction with an aperiodic lattice.

Reply 3.2: We thank the referee for his/her recognitions of our work.

Comment 3.3: The findings of this work have a high level of significance. The overall analysis, experimental data, and presentation of the results seem sound.

Reply 3.3: We thank the referee for considering our work to be “high level of significance”. We also thank the referee for the following comments to improve the presentation of our work.

Comment 3.4: I have however a concern, whether enough evidence is given that the experimentally observed modes really are Fermi arcs. If the authors can clarify the role of the topological round-trip eigenphase (see comments below), that might resolve this problem.

Reply 3.4: We thank the referee for the nice question.

In brief, the topological interface states are the planar guided modes sandwiched between the two Weyl meta-crystals. Therefore, we use the round-trip eigen phase condition (or guided mode condition) $\varphi^I + \varphi^{II} = 2n\pi$ to predict the interface Fermi arcs.

As schematically shown in Fig. R11a, the twisted bi-block Weyl meta-crystal consists of three parts: top Weyl meta-crystal, dielectric spacer layer and bottom Weyl meta-crystal. The modes that can propagate in the spacer layer are the planar guided modes satisfying the condition $\varphi = \varphi^I + \varphi^{II} = 2n\pi$, where the topologically nontrivial phase φ^I provided by reflection of Weyl meta-crystals, and trivial smooth phase φ^{II} given by propagating in spacer layer. Specifically, the guided mode is described by the product of two scattering matrices $D = \tilde{S}_{22}^{top}(\vec{k})\tilde{S}_{11}^{bot}(\vec{k})$ (see Fig. R11a), which is a 2×2 unitary matrix with each eigenvalue taking the form of $e^{i\varphi}$, that are the round-trip eigen phases. When we gradually decrease the height of spacer layer ($\varphi^{II} \rightarrow 0$), the planar guided modes finally turn to be interface modes, that are Fermi arc states (see Fig. R11b). And thus, we apply the planar guided mode condition to predict the interface Fermi arcs. Especially, when we focus on the Fermi arcs at Weyl frequency, both the top and bottom Weyl meta-crystals exhibit total reflection. Around each Weyl point the round-trip eigen phases φ are topologically nontrivial like double-sheet Riemann surface (see Fig. R11c), where Weyl points serve as singularities. In Fig. R11d we show a zoom-in view of the double-sheet Riemann surface, where the gray plane indicates the guided mode condition (interface Fermi arcs). Here, “double-sheet” is due to the two polarization states (e.g., TE and TM) of electromagnetic waves. The guided mode

condition $\varphi = 2n\pi$ is also indicated by the dashed horizontal lines in Fig. R11e, which shows the round-trip eigen phases along the encircled loop in Fig. R11b. In Fig. R11e, for an arbitrary horizontal cut of the round-trip eigen phases, i.e. considering any extra trivial phase φ^{II} , there must be one point being crossed, which indicates that there must be one Fermi arc. Similarly, any mechanism that individually changes φ^I or φ^{II} leads to the reconstruction of Fermi arcs. In our work, the twisting angle θ reshapes φ^I and thus enables continuous twisted Fermi arcs. The round-trip eigen phases actually exhibit the topology characteristic of Weyl points, we further map it onto a torus knot as shown in Fig. R11f.

In the revised main text, we have added the above elaborate explanation. We also have replotted Fig. 2 (copied as Fig. R11 below).

Fig. R11 (Fig. 2). Twisted Weyl metamaterials: Round-trip eigen phases and twisted Fermi arcs.

a, Round-trip eigen phases and planar waveguide consisting of the top/bottom Weyl meta-crystal and the middle spacer layer. **b**, Interface Fermi arcs when $\theta = 30^\circ$. **c**, The round-trip eigen phase exhibits double-sheet Riemann surface structure. **d**, The zoom-in view of the double-sheet Riemann surface structure in (c) around one Weyl point as encircled in (b). **e**, The round-trip eigen phases around one Weyl point as indicated by the dashed circle in (b), which also corresponds to the helical boundary in (d). The horizontal dashed line denotes the guided mode condition $\varphi = \varphi^I + \varphi^{II} = 2n\pi$. **f**, Torus knots built by mapping the round-trip eigen phases onto a torus.

In Fig. R12a, we show the twisted Fermi arcs directly calculated from boundary condition matching method, which are exactly the same as those predicted from the round-trip eigen phase condition as shown in Fig. R12b. Therefore, the round-trip eigen phase condition can well describe interface Fermi arcs.

Fig. R12. **a**, Interface Fermi arcs calculated by directly matching boundary condition. **b**, Interface Fermi arcs predicted by the round trip eigen phase condition.

When the twisting angle is 0, the top and bottom Weyl meta-crystals are a whole block, i.e. a single Weyl meta-crystal. At Weyl point frequency, we scan the interface and there four isolated Weyl points, as shown in the Fig. R13a. When the twisting angle is 30 degrees, two pairs of bright arcs

emerge in momentum space as shown in Fig. R13b, where their endpoints coincide with the Weyl points, that are Fermi arcs.

Fig. R13. Experimental evidence of twisted Fermi arcs. **a**, Four isolated Weyl points are probed when $\theta = 0$ at 6.9 GHz. **b**, Fermi arcs bridging top and bottom Weyl points when $\theta = 30^\circ$.

Comment 3.5: If the authors can address this and the other points listed below satisfactorily, I would support the publication of the manuscript.

Reply 3.5: We thank the referee for supporting the publication of our work.

Comment 3.6: Here the detailed comments and questions, labeled by the line numbers of the manuscript:

34: “Singular reflection phase” is unclear at that point.

Reply 3.6: We apologize for the unclear expression.

Here, “singular reflection phase” refers to the topologically nontriviality of round-trip reflection phase around each Weyl point. The round-trip phase consists of a double-sheet Riemann surface as shown in Fig. R11d, and each Weyl point serves as the singularity of the Riemann surface.

For more details, please see **Reply 3.4**.

Comment 3.7: 41-43: Clarify that translation invariance in the plane parallel to the interface is meant. (If I understand it correctly).

Reply 3.7: In the main text, we mentioned that “Most previous studies focus on the system preserving translation symmetry as Weyl points can be gapped by backward scattering.”.

For the structures preserving translational symmetry, both Weyl points and Fermi arcs could be well described by usual Bloch band theory. Nevertheless, in the cases that are absence of translational symmetry, Bloch bands are ill-defined. Here, in the twisted bi-block photonic Weyl meta-crystals the interface exhibits aperiodic features and lacks of translation invariance, which makes solving Fermi arcs much more challenging. Our work proposes the round-trip eigen phases and find interface Fermi arcs without translation invariance. In a word, translation invariance is very important for conventional Fermi arcs, while lacking translation symmetry there still existing Fermi arcs but exhibiting much more complex distribution.

Comment 3.8: 45-46: What is the significance of broken translation invariance with regard to applications and what are such applications?

Reply 3.8: Thank the referee for the nice question.

Firstly, the majority of natural systems lacks translational symmetry, i.e. quasicrystals and even disorder. The study of incommensurate lattices (broken translation invariance) could generalize those physics that were only believed to be stable under translation invariance. Weyl points and Fermi arcs belong to such systems and are both defined under translation symmetry, the

generalization is fundamentally important. Here, we apply round-trip eigen phases to study interface Fermi arcs and show their incommensurate distribution in the momentum space. The conclusion works for all other systems, including electronic, photonic, acoustics, mechanics, etc. Secondly, systems with broken translation symmetry have unique electronic, optical and mechanical properties, for example quasicrystals. In Weyl photonics, the spatial arrangement of Weyl points in incommensurate lattices could help in manipulating interfacial electromagnetic waves so as to continuously twist their propagation directions. Especially, the equi-frequency contour takes the form of arcs, which means highly anisotropic and directional. In the revised turn, we have added the above argument in the discussion section.

Comment 3.9: 51: The phrase “interaction between topology and physics” is unclear.

Reply 3.9: We apologize for the unclear phrase.

Very recently, there is an upsurge in the application of vertically stacked two-dimensional material structures. One of the widely used approach to manipulate the topological physics is to generating Moiré pattern by introducing interlayer mismatch, through stacking and twisting. Here, the phrase “interaction between topology and physics” actually indicates the possible exotic topological physics under different stacking and twisting configurations.

In the revised main text, we have replaced it with “topological physics” to avoid any confusion.

Comment 3.10: 95-97: The terminology Weyl meta-materials and meta-crystals needs to be explained better at this point. I understood what is meant only after reading the sentences in lines 155-158.

Reply 3.10: We apologize for the missing terminology definitions.

In our manuscript, metamaterials refer to homogenous medium, which could be described by electromagnetic constitutive relations. While the term meta-crystal particularly indicates the photonic crystal that is constructed by metamaterials, i.e., periodically distributed metamaterials in spatial space. In the revised main text, we have added several definitions for the two terminologies when mentioned.

Comment 3.11: 111: What exactly is the round-trip eigenphase? How exactly are the round-trip eigenphases used here? Are they useful to identify the Fermi arcs theoretically or are they used to identify the Fermi arcs experimentally?

Reply 3.11: We thank the referee for the nice question.

We have elaborately explained round-trip eigen phases above, please refer to **Reply 3.4**.

The twisted bi-block Weyl meta-crystal consists of three parts: top Weyl meta-crystal, dielectric spacer and bottom Weyl meta-crystal. The modes that can propagate in the spacer layer are the planar guided modes. When we gradually decrease the height of the spacer layer, the planar guided modes finally turn to be interface modes, that are Fermi arc states. And thus, we apply the planar guided mode condition to predict the interface Fermi arcs. A guided mode can exist only when the transverse resonance condition is satisfied, i.e. the repeatedly reflected wave has constructive interference with itself. Thus, guided mode must be an eigenmode of the round trip. For each

eigenmode, its state (or polarization) keeps still after a round-trip, the only freedom left is the accumulated phase, dubbed as eigen phase here. The round-trip eigen phases φ are used to characterize the planar guided mode condition with $\varphi = 2n\pi$. In other words, when the round-trip eigen-phases satisfying $\varphi = 2n\pi$, we theoretically identify the positions of Fermi arcs in the momentum space.

Comment 3.12: 118: In Fig 2b,c it looks like a phase of 2π is accumulated by going around the Weyl node **twice**. This appears contradictory to what is stated in the text.

Reply 3.12: Thank you for pointing this out.

Here, we show the round-trip eigen phases around one Weyl point, which are not the accumulated phase (such as Berry phase for a picked state) going around the Weyl node in momentum space. Round-trip eigen phase indicates the phase accumulated by one eigenmode after travel a round-trip in real space (please see Fig. 2a). In photonics or electromagnetic systems, there are two modes/freedoms (transverse electric TE and transverse magnetic TM modes) for each propagating wavevector \vec{k} . In other words, we have two round-trip eigenmodes for each \vec{k} and their eigen phases consist of a double-sheet Riemann surface around one Weyl node (please refer to Fig. 2 or **Reply 3.4**). If for each \vec{k} , there are N eigenmodes, their eigen phases will finally construct a N -sheet Riemann surface, which is determined by the internal freedoms.

Comment 3.13: 205: What are the “up and down cycles” and why are they necessary for the scanning? Fig 4b seems to be not very useful in the description.

Reply 3.13: We thank the referee for pointing this out.

The “up and down cycles” in this work refers to the path the probe antenna travelled for probing the interface Fermi arcs. In order to probe electromagnetic waves on the interface sandwiched between two Weyl meta-crystal, we have to reach the interface. We periodically drilled holes at the center of each unit cell to ensure the interface wave probing. We use a XYZ stage that controls the probe antenna to move along the up and down cycles. Only if the probe antenna reaches the interface, the signal collection of VNA (vector network analyzer) is triggered (as highlighted by red points in Fig. R14b).

In the revised turn, we have updated Figs. 4a and b as copied below for convenience.

Fig. R14 (Figs. 4a-b). **a**, Schematic experimental setup of the twisted bi-block Weyl meta-crystal. The sample has been truncated into a cylinder shape for the continuous twisting purpose. Two portions have been cut to show the source and probe positions. The source antenna is inserted through the central hole of the bottom Weyl meta-crystal. **b**, Schematic cutting view of the probe scanning path, where the probe antenna raster-scans the interface following a cycle manner ($P_1 \rightarrow P_2 \rightarrow P_1 \dots$).

Comment 3.14: 227: Fig. 4 a and b caption is too short. At least a hint could be given that the setup is described in the text.

Reply 3.14: Thank you for pointing this out.

In the revised turn, we have added a method section in the main text to elaborately describe the interface scanning details. We also have added the following text in the caption of Figs. 4a and b.

“Figure 4. a, Schematic experimental setup of the twisted bi-block Weyl meta-crystal. The sample has been truncated into a cylinder shape for the continuous twisting purpose. Two portions have been cut to show the source and probe positions. The source antenna is inserted through the central hole of the bottom Weyl meta-crystal. b, Schematic cutting view of the probe scanning path, where the probe antenna raster-scans the interface following a cycle manner ($P_1 \rightarrow P_2 \rightarrow P_1 \dots$).”.

Comment 3.15: 228: With regard to the strong dependence of the arcs on the fitting parameter ϕ' , shown in Fig. 3, please explain the value of the experiment - theory correspondence.

Reply 3.15: We thank the referee for the nice question.

As mentioned in the main text, ϕ^{II} is a trivial phase without any singularities (for brevity, we have made the replacement $\phi' \rightarrow \phi^{II}$). The presence of ϕ^{II} does not alter the topology of Fermi arcs globally, even though Fermi arcs may undergo deformation locally with varying ϕ^{II} (For a very relative example, please refer to Fig. S4.).

In our theoretical RCWA method, we construct the Weyl meta-crystal via spatially distribute Weyl metamaterials, which is derived from the underlying electromagnetic responses of the saddle shaped metallic structure. Although the theoretical model shares the same symmetry and Weyl node configuration with the realistic sample, they lack of the detailed geometries. In other words, the theoretical model is a qualitative analysis and we need change parameter ϕ^{II} to fitting the

experiment. However, since φ^{II} does not change the topology, we can always find one Fermi arc patterns that fit well with the experiment results.

Reviewer #1 (Remarks to the Author):

The authors have adequately responded to all the queries and recommendations in my initial report. However, I believe there is room for additional enhancements to the manuscript before it can be accepted in principle in Nature Communication. These improvements can enhance the manuscript's clarity for a broader audience. Below are the specific points that need to be addressed:

1) In the text (lines 69-70), it is stated that for small angles, the meta-crystal can be regarded as a homogenous medium (i.e., metamaterials) when the quasi-period is much larger than the Weyl wavelength (the wavelength at the Weyl frequency). However, in the subsequent figures that refer to metamaterials, the angle is not small. This happens, for example, in Fig. 3 with a twist angle ranging from 30° to 56.55° . Meanwhile, in Fig. 5, with angles ranging from 15° to 75° , they refer to meta-crystal!

Could the authors better clarify the notion of metamaterial and meta-crystal regimes in the text?

2) Fixing the choice of units of the angle has yet to be systematically performed throughout the figures of the manuscript. After reading this newer version, I have concluded that it would be better to express the twist angles in degree (close to human understanding) and the angles due to the round-trip eigenphases in radians (more related to the $2n\pi$ condition). The authors should address this point in the text and the figures.

3) Since the manuscript must be scientifically accurate and understandable, the authors should avoid jargon used when doing research. For instance, the term "twisted Fermi arcs" here refers to the interface Fermi arcs between two Weyl semimetals rotated from each other; the Fermi arcs are not twisted as they have been reconstructed. The words "twisted Weyl semimetals," "twisted Weyl meta-crystals," etc., are not precisely correct: one might say "twisted bi-block..." here, but a twisted Weyl semimetal refers to something else. The text should be improved consequently.

4) In line 71, the "Weyl wavelength" is defined by giving an equivalent definition, "Weyl frequency," which I suppose is the electromagnetic wavelength/frequency of the Weyl points. The text should be improved consequently.

Reviewer #2 (Remarks to the Author):

I have gone through the authors' replies to the referees' comments and the changes made to the manuscript and found them to be satisfactory. The authors have added clarifications and details about the physical setup and have also modified the figures to make them clearer. Additionally, they have performed further experiments that convincingly answer my questions.

Overall, I believe that this manuscript explores the interesting interface phenomena between two Weyl semimetal phases, including the incommensurate case. As was recently argued in arXiv:2304.13118, interfaces between topological systems hold many surprises and thus deserve further attention. I can therefore happily recommend this manuscript for publication in Nature Communications.

I have two suggestions:

1. "Before Hybridize" in figure 1 should be changed to "Before Hybridization".

2. The authors may wish to cite some references from the large body of experimental work on photonic Weyl points, as I found these to be missing from the introduction. This will help readers place the present work in the right context: For example, see Science, 359(6379), pp.1013-1016 (2018), Phys. Rev. X 7, 031032 (2017), Nature Physics volume 13, pages 611–617 (2017), Phys. Rev. Lett. 125, 253902 (2020), Phys. Rev. Lett. 125, 143001 (2020), Laser & Photonics Reviews 16, 2100452

(2022).

Reviewer #3 (Remarks to the Author):

Overall, the authors addressed the criticism to my satisfaction. I only have two minor comments:

1. A comment related to newly added text on transport properties of interface Fermi arcs (line 41) and the observed Fermi-arc Lifshitz transitions:

In electronic systems, nearly crossing interface Fermi arcs, i.e., close to a Lifshitz transition, have been predicted to

lead to magnetic-breakdown effects in Phys. Rev. B 107, L241109 (2023) and arXiv:2310.19720. The authors might want to cite these works (e.g. in line 41 or/and when discussing the Lifshitz transitions).

2. Related to 3.8: The newly added arguments in the discussion section in response to the comment 3.8 are enlightening. When speaking about the generalization of topological physics of Weyl semimetals to broken translation invariance, some literature addressing this problem could be cited, such as on topological amorphous metals, Phys. Rev. Lett. 123, 076401 (2019), and Fermi-arc metals, Phys. Rev. Lett. 130, 196602 (2023).

RESPONSE TO REVIEWERS

Reviewer #1 (Remarks to the Author):

Comment 1.1: The authors have adequately responded to all the queries and recommendations in my initial report. However, I believe there is room for additional enhancements to the manuscript before it can be accepted in principle in Nature Communication. These improvements can enhance the manuscript's clarity for a broader audience. Below are the specific points that need to be addressed:

1) In the text (lines 69-70), it is stated that for small angles, the meta-crystal can be regarded as a homogenous medium (i.e., metamaterials) when the quasi-period is much larger than the Weyl wavelength (the wavelength at the Weyl frequency). However, in the subsequent figures that refer to metamaterials, the angle is not small. This happens, for example, in Fig. 3 with a twist angle ranging from 30° to 56.55° . Meanwhile, in Fig. 5, with angles ranging from 15° to 75° , they refer to meta-crystal! Could the authors better clarify the notion of metamaterial and meta-crystal regimes in the text?

Reply 1.1: We thank the reviewer for the additional constructive and insightful suggestions to improve the clarity of our work.

Metamaterials imply homogeneous media, i.e., there is no spatial-periodic modulation. From metamaterials, we can calculate and predict twisted Fermi arcs for each arbitrary twisting angle. The results are mainly shown in Figs. 2 and 3. However, it is not experimentally realistic. In other words, the realistic structure is periodically

distributed in the spatial space. We have to apply meta-crystals to describe the experimental results. Well, the meta-crystal theory is generalized from metamaterials, which coincides with that of metamaterials at small twisting angles, e.g. $\theta \leq 30^\circ$. Therefore, in Fig. 5 we apply meta-crystal results for all twisting angles ranging from 15 to 75 degrees.

In the caption of Figs. 2 to 5, we added “Twisted bi-block Weyl metamaterials” and “Twisted bi-block Weyl meta-crystals”, respectively, to indicate the different theories used.

In the revised turn, we have added the following paragraph to avoid any confusion, **“We first show the way to solve interface Fermi arcs of homogeneous Weyl metamaterials (see section I in Supplementary Information and Supplementary Fig. 1) and then consider the periodic modulation in Weyl meta-crystals. Metamaterials imply homogeneous media, i.e., there is no spatial-periodic modulation. From metamaterials, we can calculate and predict twisted Fermi arcs for each arbitrary twisting angle. However, it is not experimentally realistic. In other words, the realistic structure is periodically distributed in the spatial space. We have to apply meta-crystals to describe the experimental results.”.**

Comment 1.2:2) Fixing the choice of units of the angle has yet to be systematically performed throughout the figures of the manuscript. After reading this newer version,

I have concluded that it would be better to express the twist angles in degree (close to human understanding) and the angles due to the round-trip eigenphases in radians (more related to the $2n\pi$ condition). The authors should address this point in the text and the figures.

Reply 1.2: We thank the reviewer for pointing this out. We have unified the use of angles and radians in the revised turn, that twisted angles are expressed in degrees, and angles due to the round-trip eigen phase in radians.

Comment 1.3:3) Since the manuscript must be scientifically accurate and understandable, the authors should avoid jargon used when doing research. For instance, the term “twisted Fermi arcs” here refers to the interface Fermi arcs between two Weyl semimetals rotated from each other; the Fermi arcs are not twisted as they have been reconstructed. The words “twisted Weyl semimetals,” “twisted Weyl meta-crystals,” etc., are not precisely correct: one might say “twisted bi-block...” here, but a twisted Weyl semimetal refers to something else. The text should be improved consequently.

Reply 1.3: We thank the reviewer for pointing this out.

In the revised turn, we have defined “twisted Fermi arcs” in the work following the reviewer’s clear explanation as,

“the twisted Fermi arcs (refers to the interface Fermi arcs between two Weyl semimetals rotated from each other)”.

We also have updated all the “twisted bi-block Weyl metamaterials/meta-crystals”.

Comment 1.4:4) In line 71, the “Weyl wavelength” is defined by giving an equivalent definition, “Weyl frequency,” which I suppose is the electromagnetic wavelength/frequency of the Weyl points. The text should be improved consequently.

Reply 1.4: We thank the reviewer for the very nice suggestion. In the revised manuscript, we have updated the related phrases.

Reviewer #2 (Remarks to the Author):

Comment 2.1: I have gone through the authors' replies to the referees' comments and the changes made to the manuscript and found them to be satisfactory. The authors have added clarifications and details about the physical setup and have also modified the figures to make them clearer. Additionally, they have performed further experiments that convincingly answer my questions.

Overall, I believe that this manuscript explores the interesting interface phenomena between two Weyl semimetal phases, including the incommensurate case. As was recently argued in arXiv:2304.13118, interfaces between topological systems hold many surprises and thus deserve further attention. I can therefore happily recommend this manuscript for publication in Nature Communications.

Reply 2.1: We thank the reviewer for accepting our reply and recommending our work for publication in Nature Communications.

Comment 2.2: I have two suggestions:

1. "Before Hybridize" in figure 1 should be changed to "Before Hybridization".

Reply 2.2: We thank the reviewer for pointing this out. In the revised manuscript, we have updated it.

Comment 2.3:2. The authors may wish to cite some references from the large body of experimental work on photonic Weyl points, as I found these to be missing from the introduction. This will help readers place the present work in the right context: For

example, see Science, 359(6379), pp.1013-1016 (2018), Phys. Rev. X 7, 031032 (2017), Nature Physics volume 13, pages 611–617 (2017), Phys. Rev. Lett. 125, 253902 (2020), Phys. Rev. Lett. 125, 143001 (2020), Laser & Photonics Reviews 16, 2100452 (2022).

Reply 2.3: We sincerely thank the reviewer for bringing to our notice these very related works. In the revised main text, we have cited them in the introduction part.

Reviewer #3 (Remarks to the Author):

Comment 3.1: Overall, the authors addressed the criticism to my satisfaction. I only have two minor comments:

1. A comment related to newly added text on transport properties of interface Fermi arcs (line 41) and the observed Fermi-arc Lifshitz transitions: In electronic systems, nearly crossing interface Fermi arcs, i.e., close to a Lifshitz transition, have been predicted to lead to magnetic-breakdown effects in Phys. Rev. B 107, L241109 (2023) and arXiv:2310.19720. The authors might want to cite these works (e.g. in line 41 or/and when discussing the Lifshitz transitions).

Reply 3.1: We thank the reviewer for accepting our reply. We also sincerely thank the reviewer for bringing to our notice these very related works. In the revised main text, we have cited them.

Comment 3.2: 2. Related to 3.8: The newly added arguments in the discussion section in response to the comment 3.8 are enlightening. When speaking about the generalization of topological physics of Weyl semimetals to broken translation invariance, some literature addressing this problem could be cited, such as on topological amorphous metals, Phys. Rev. Lett. 123, 076401 (2019), and Fermi-arc metals, Phys. Rev. Lett. 130, 196602 (2023).

Reply 3.2: We thank the reviewer for his/her positive assessment of our discussion. In the revised main text, we have cited them.